# The Denaturant- and Mutation-Induced Disassembly of *Pseudomonas aeruginosa* Hexameric Hfq Y55W Mutant

**DOI:** 10.3390/molecules27123821

**Published:** 2022-06-14

**Authors:** Victor Marchenkov, Natalia Lekontseva, Natalia Marchenko, Ivan Kashparov, Victoriia Murina, Alexey Nikulin, Vladimir Filimonov, Gennady Semisotnov

**Affiliations:** Institute of Protein Research, Russian Academy of Sciences, Institutskaya Street 4, 142290 Pushchino, Russia; march@phys.protres.ru (V.M.); lekontseva@vega.protres.ru (N.L.); lita@phys.protres.ru (N.M.); ivkashp@vega.protres.ru (I.K.); victoriia.murina@gmail.com (V.M.); nikulin@vega.protres.ru (A.N.)

**Keywords:** Hfq hexamer, mutations, unfolding intermediates, fluorescence, thermodynamics

## Abstract

Although oligomeric proteins are predominant in cells, their folding is poorly studied at present. This work is focused on the denaturant- and mutation-induced disassembly of the hexameric mutant Y55W of the Qβ host factor (Hfq) from mesophilic *Pseudomonas aeruginosa* (*Pae*). Using intrinsic tryptophan fluorescence, dynamic light scattering (DLS), and high-performance liquid chromatography (HPLC), we show that the dissociation of Hfq Y55W occurs either under the effect of GuHCl or during the pre-denaturing transition, when the protein concentration is decreased, with both events proceeding through the accumulation of stable intermediate states. With an extremely low pH of 1.4, a low ionic strength, and decreasing protein concentration, the accumulated trimers and dimers turn into monomers. Also, we report on the structural features of monomeric Hfq resulting from a triple mutation (D9A/V43R/Y55W) within the inter-subunit surface of the protein. This globular and rigidly packed monomer displays a high thermostability and an oligomer-like content of the secondary structure, although its urea resistance is much lower.

## 1. Introduction

The mysterious puzzle of how protein chains, initially disordered, form their spatial structure has not yet been solved. While the main folding steps of relatively small globular proteins are generally known [1,2,3,4], those of large and oligomeric proteins remain unclear [4,5,6,7]. One of these issues is the relationship between the protein’s quaternary structure, its stability, and folding.

This study is focused on the hexameric protein Hfq from *Pseudomonas aeruginosa* that acts as the main mediator in the regulatory network of gene expression by small RNAs [8,9]. It belongs to the Sm/LSm superfamily of proteins with a ring-like quaternary structure containing six identical subunits of about 10 kDa each [10]. There are some common properties of Hfqs from various organisms. One of the properties is the increased thermostability of Hfq, even from mesophilic bacteria [11,12]. This property, for example, greatly simplifies the purification process of these proteins from cellular lysates, i.e., it requires only heating to 80 °C and then separating the aggregates of contaminating proteins by centrifugation [12,13,14]. The second common property of Hfq proteins is their ring-like quaternary structure that provides the binding of single-stranded RNA [12] via multiple sites on the protein surface. Thus, the study of the stability of the Hfq quaternary structure is important to elucidate the protein function [15]. The Hfq protein from mesophilic *Pseudomonas aeruginosa* displays super-thermostability, with a half-transition temperature at neutral pH and about 116 °C [11,16]. Previously, we have shown that the substitution of Ala for the conserved residues Gln8, Asp40, and Tyr55 results in a decrease of Hfq thermostability, while its hexameric quaternary structure remains unchanged [16]. A two-fold decrease of the protein concentration does not affect the main differential scanning calorimetry (DSC) peak but visibly influences its left shoulder [16]. This observation allows us to propose a three-state model of Hfq thermal unfolding.

Here, we verify this suggestion by equilibrium denaturation studies of the protein, which allow us to obtain information about its conformational stability [17,18]. In thermodynamic experiments on oligomeric proteins, their concentration was used as an additional regulator of equilibrium [15,19,20,21,22]. These experiments require techniques other than differential scanning calorimetry (DSC) or circular dichroism (CD) which are efficient within a wide protein concentration range. In DSC, the lower limit of protein concentration is about 10 μM, while its ten-fold increase can provoke protein aggregation during thermal unfolding. The CD usage is often hampered by a low signal amplitude at low protein concentrations. Unlike these methods, fluorescence works within a sufficiently broad range of protein concentrations [18,21]. Because the wild type (WT) Hfq lacks tryptophan (Trp) [14], which is the most effective natural fluorescence probe, we replaced a tyrosine residue within the inter-subunit contact (Y55) with tryptophan. A preliminary test of the Hfq structure by the Swiss-Model program [23,24] showed no serious distortions resulting from this substitution. The expression of the plasmid encoding this mutant in *Escherichia coli* (*E. coli*) gave a product with a hexameric conformation close to that reported for the WT protein [14], as shown by standard biochemical tests and crystallographic studies.

In addition to substitutions in the Hfq sequence at position H57 and some other positions (e.g., Q8A) analyzed previously [11,16], here, we report on the single (Y55W) and triple (Y55W/D9A/V43R) Hfq mutants. The aim of introducing a tryptophan residue into the inter-subunit area is to use fluorescence in examining the denaturant-induced Hfq Y55W disassembly. We replaced D9 with Ala for two reasons: firstly, to avoid the spontaneous hydrolysis of the peptide bond, D9-P10 [25], and secondly, to remove a negative charge compensating, to some extent, for the positive charge of the conserved K56 protected from solvent within the inter-subunit contacts [14]. It turned out that single substitutions do not noticeably change the hexameric Hfq structure at neutral pH, but rather, decrease its stability. However, an additional bulky positive charge introduced into the inter-subunit area through the V43R substitution resulted in both inhibition of the hexamer formation and biosynthesis of the folded monomer. Note that V43R mutation of *Escherichia coli* Hfq results in a very negative effect on the protein function [26].

## 2. Results

In proteins, the intrinsic probe most sensitive to its surrounding is a tryptophan residue (Trp) [27,28]. Because the WT Hfq does not contain Trp, here, Y55 belonging to the protein inter-subunit surface was substituted for Trp. To obtain the monomeric Hfq with Trp fluorescence, two additional mutations, D9A and V43R, were introduced.

### 2.1. Structural Properties of the Hexameric Single (Y55W) and Monomeric Triple (Y55W/D9A/V43R) Hfq Mutants

The three-dimensional crystal structure of the Hfq Y55W mutant, deposited in the PDB under the code 5I21, is presented in Figure 1; this confirms that the mutation Y55W does not change the overall hexameric structure of the protein (see also [14]). Yet, some changes must be mentioned. Figure 1 presents the superposition of the main chain conformations for one subunit of WT [14] and Y55W variants of Hfq. Small deviations are observed in the loop positions, while all elements of the mutant secondary structure remain intact. Moreover, the position of the tryptophan indole ring is slightly different from that of the benzene one for intact tyrosine 55, possibly due to the bulky side group being included. As Y55 engages in stacking-like interactions with conserved H57 [14], this local distortion may influence the protein stability.

Figure 2 presents some structural properties of the Hfq hexameric Y55W and monomeric Y55W/D9A/V43R mutants. First, the electrophoretic mobility of the monomer is much higher than that of the hexamer (Figure 2a), while its presence as a monomer in solution was confirmed by cross-linking with a glutaric aldehyde (not shown). Second, despite the similarity in the secondary structures of spatial crystal structures of WT Hfq and its Y55W mutant (Figure 1), there is a distinct difference in their far-UV CD spectra (Figure 2b). This difference may be attributed to either some destabilization of the Hfq Y55W secondary structure in solution or the optical properties of the tryptophan residue inserted into its current surrounding [29], apparently reflecting its interaction with the nearest neighbors within the monomer (subunit). At the same time, the deconvolution of the CD spectra of WT Hfq, hexameric, and monomeric mutants (using one of the latest programs available on the internet [30]) shows the similarity of the secondary structure content of these forms (Table 1).

Third, intrinsic Trp fluorescence spectra (Figure 2c) indicate that in monomeric Hfq variants, Trp is more exposed to the solvent than in the hexamer (the spectrum is shifted to the long-wavelength region), but less exposed than in the unfolded state. Thus, the Trp fluorescence spectrum position is a good indicator of the disassembly of the protein’s oligomeric structure.

Figure 3 shows the resistance of hexameric and monomeric Hfq mutants against urea, GuHCl, and temperature. Transverse urea-gradient electrophoresis (TUGE) allowed us to visualize the urea-induced denaturation of globular proteins caused by a change in the hydrodynamic size of the protein chain [31]. As seen in Figure 3a, hexameric Hfq Y55W retains its hydrodynamic size (i.e., does not unfold) under the effect of urea. The hexameric WT Hfq displays the same behavior toward urea (not shown). In contrast, the Hfq monomeric triple mutant undergoes an S-like urea-induced transition that results in an increased hydrodynamic size (decreased electrophoretic mobility) during the unfolding (Figure 3b). This transition is completely reversible, as confirmed by monitoring the protein refolding with transverse urea-gradient gel electrophoresis (not shown). This observation proves that the triple mutant has a rigidly packed tertiary structure with the thermodynamic parameters of its unfolding (shown in Table 2), which are typical for globular proteins of similar size [32]. Interestingly, despite the monomeric triple Hfq mutant showing low urea resistance in comparison with the monomer within the hexameric HfqY55W (Figure 3a,b), it retained relatively high thermostability. The CD spectrum of the triple mutant recorded at 90 °C (Figure 3d) showed a high content of secondary structure elements (α = 10.2%, β = 28.1%, other = 61.7%), i.e., similar to that at 25 °C (data in Table 1).

A stronger denaturant, GuHCl, causes the unfolding of both hexameric WT Hfq and its Y55W mutant (Figure 3c and Appendix A). The GuHCl-induced changes in CD spectra for WT Hfq occur within a much wider range of denaturant concentrations with much less parameter m than that expected for the cooperative two-state unfolding of globular proteins with a molecular weight of about 55 kDa [32] (see also [11]), thus unequivocally demonstrating an alternative non-two-state type of the WT Hfq unfolding transition. The substitution Y55W essentially decreases the resistance of the Hfq hexamer against GuHCl, and its unfolding transition is apparently more cooperative (Figure 3c) than in the case of WT Hfq, resembling a simple two-state transition. However, as seen in Figure 3c, an eleven-fold decrease of Hfq Y55W concentration resulted in a shift to the left of the part of the unfolding curve, as would be expected from a process involving the dissociation of an oligomer. Therefore, the hexameric protein unfolding data were analyzed using the three-state model (see Materials and Methods). Initially, the deconvolution procedure based on the three-state model (Equations (2) and (17)) was applied to the protein denaturing transition monitored by fluorescence spectra decomposition, as described below, with parameters as shown in Table 2. Afterward, the three-state model (with the fixed values of the m and *K* parameters for the dissociation and unfolding processes) was applied to the fitting of the protein unfolding transitions monitored by Far-UV CD (Figure 3c).

### 2.2. The Intermediate Species Accumulated during GuHCl-Induced Denaturation of the Hexameric Hfq Y55W

As seen from Figure 3c, an eleven-fold decrease in Hfq Y55W concentration resulted in a change of the GuHCl-induced unfolding, as monitored by far-UV CD within the 1.6 M–2.2 M denaturant concentration range. With this in mind, we selected a GuHCl concentration, providing a maximally populated native state of the protein with a minimally populated unfolded state. Varying protein concentrations, we analyzed the unfolding transition depending on the protein concentration. For this purpose, we used the Trp fluorescence spectra position (center of mass (CM), Equation (1)) which is mainly sensitive to hexamer dissociation (see Figure 2c) and can be registered with a reasonable noise level at protein (monomer) concentrations as low as to 0.1 μM.

Figure 4a demonstrates a set of Hfq Y55W Trp fluorescence spectra recorded within the 0–5 M GuHCl range at three protein (monomer) concentrations: 0.8 (top), 6.5 (middle), and 65 µM (bottom). These spectra within intermediate concentrations of GuHCl apparently are at least two-component.

Figure 4b represents the dependence of the spectrum center of mass on the GuHCl concentration at different protein concentrations. The standard fluorescence spectra (Figure 2c) were used for the deconvolution of Hfq Y55W fluorescence spectra during GuHCl-induced protein unfolding at different protein concentrations. An example of such deconvolution is presented in Figure 4d. The result of such a deconvolution for the protein unfolding transitions at three protein concentrations is shown in Figure 4c. An important conclusion derived from these data is that the lower the protein concentration, the higher the accumulation level of the intermediate state fraction. Thus, at a high protein concentration, the denaturation mode is apparently close to the two-state one. This signifies that the stability of the protein inter-subunit interactions approaches that of the monomer (subunit) at high protein concentrations.

### 2.3. Intermediate Species Accumulated during Protein Concentration Decrease

The concentration-dependent dissociation of the oligomeric protein structure and the nature of the accumulated intermediate state were determined from the protein concentration dependence (at fixed GuHCl concentration) of the protein fluorescence spectra position and the hydrodynamic size. The GuHCl concentration was 1.6 M, providing the maximally populated intermediate at low protein concentrations (see Figure 4c).

Figure 5a presents the set of Hfq Y55W fluorescence spectra at various protein concentrations at 1.6 M GuHCl. The position of the maximum of the protein spectra recorded below 1 μM protein concentration (inset of the Figure 5a) is close to that for the monomeric (Y55W/D10A/V43R) spectrum (Figure 2c).

Figure 5b shows the dependence of state fractions (N and I) on the molar Hfq Y55W (monomer) concentration at 1.6 M GuHCl, as determined by fluorescence spectra deconvolution using the standard spectra of hexamer and monomer presented in Figure 2c.

Size exclusion chromatography was used to determine the nature of the intermediate state accumulated during a decrease in protein concentration. Figure 6a presents several standardized elution profiles of Hfq Y55W at 1.6 M GuHCl with decreasing concentrations of loaded protein, as monitored by tryptophan fluorescence.

As seen, there are two major and one minor elution peaks. The left minor peak evidently corresponds to some associates and disappears with decreasing protein concentration. The other left major peak shows hexamer elution (Figure 6a,b) and does not change its position with decreasing protein concentration, although its amplitude is diminished (Figure 6a,c). The right peak shows dimer elution (Figure 6a,b) at high protein concentrations, while at low ones, it is close to a monomer (Figure 6b). Similar behavior was reported for the dimeric factor for inversion stimulation [22,33] and may be interpreted as a protein concentration-dependent exchange between the dimeric and monomeric forms of the protein.

### 2.4. Intermediate Species Accumulated at Low pH

The dissociation of hexameric Hfq Y55W occurs at low pH. Figure 7a presents a selected set of Trp fluorescence spectra at various pH, from 6.0 up to 1.4, and a protein (monomer) concentration of 67 μM. With a pH as low as 4.0, the fluorescence intensity was strongly quenched without a peak shift. By further decreasing pH, the fluorescence intensity increased, and the spectrum exhibited a red shift, achieving the position between the peaks for hexameric and monomeric forms (Figure 7b). Moreover, at a pH below 4.0, the protein fluorescence spectra seem to be at least two-component, probably due to the presence of more than one species with different spectrum positions (Figure 7a). The red shift of the protein fluorescence spectrum mainly indicates the dissociation of the protein’s hexameric structure (see Figure 5). At a pH value between 4.0 and 3.0 (Figure 7a), the change in fluorescence intensity may have resulted from Trp fluorescence quenching by the nearest discharged carboxyl groups [27,28], while with a further pH decrease, the Trp fluorescence intensity increase may have been caused by diminishing the quenching due to the dissociation of the oligomeric structure. To evaluate the size of the species presented at pH 1.4, we performed DLS measurements (Figure 8a). The hexameric WT Hfq at pH 1.4 and Hfq Y55W mutant at pH 7.6 showed a symmetric peak corresponding to ~62 Å, which agrees with crystallographic data (Figure 1). At pH 1.4, the asymmetric peak for Hfq Y55W shifted toward lesser sizes. The decomposition of this peak hints at the presence of two main species with hydrodynamic sizes of ~35 Å (trimer) and ~23 Å (dimer).

To clarify the situation with the intermediates at pH 1.4, we performed size-exclusion chromatography of Hfq Y55W at pH 1.4 and various concentrations of the loaded protein. The results are presented in Figure 8b. As seen, there were three distinct peaks corresponding to at least three species.

Unfortunately, it is very difficult to estimate the exact size of these species because of ambiguous column calibration at low pH. Nevertheless, the first (left) small peak evidently corresponds to the hexamer, as was established by chromatography of WT Hfq, which does not dissociate at pH 1.4. Moreover, the elution volume of two less mobile species showed the tendency of depending on protein concentration, i.e., it shifted toward a lower hydrodynamic volume with decreasing protein concentration. Based on the results presented in Figure 7 and Figure 8, we propose that at pH 1.4 and a low ionic strength, the hexameric Hfq Y55W mutant dissociates into trimers, dimers, and monomers, which exchange with each other. The fractions of hexamers, trimers, and dimers decrease, and the population of monomers increases with decreasing protein concentration.

Thus, like the GuHCl effect (Figure 4), the repulsion of positive charges at decreasing pH destabilizes the oligomeric structure of Hfq Y55W, while the lowering protein concentration results in its dissociation into monomers.

## 3. Discussion

A major result of this study is that the high stability of the Hfq subunit leads to a non-two-state mode of hexamer unfolding. Inter-subunit interactions play an important role in maintaining the oligomer structure of the proteins, especially in cases when isolated subunits are unable to maintain their folded conformation. In these cases, the inter-subunit interactions result in the formation of a cooperative structure which unfolds in a two-state process (for example, in the case of GroES heptamer of similar to the Hfq size [34]). As we suggested previously [11,16] and confirmed here, at neutral pH, the three-state model adequately describes the GuHCl-induced unfolding of Hfq hexamer (Figure 3 and Figure 4). The relatively high population of the monomeric intermediate during hexamer unfolding indicates that the isolated subunits are stable, and makes it possible to independently analyze the hexamer dissociation by varying protein concentration at a fixed GuHCl concentration within the pre-denaturing range (Figure 5). At selected GuHCl activity, the fraction of the intermediate monomer strongly depends on protein concentration, reaching a level of 80% at the lower limit of protein concentration suitable for fluorescence measurements (Figure 4, Figure 5 and Figure 6). In contrast, at the highest protein concentration used in this study, the intermediate state population was only about 30%, and at further increase in protein concentration, the unfolding would tend to the two-state model simply because something similar to an isosbestic point at 356 nm is observed (Figure 4a). In any case, with GuHCl > 3 M, the hexamer population became negligible, and only monomer unfolding was observed (Figure 4c).

Here, we demonstrated that the isolated subunit of Hfq can not only fold independently, but can also form an extremely stable structure. *Pseudomonas aeruginosa* is not a thermophile, and there is no natural need to produce extremely stable proteins. Among possible reasons, we can mention the following: (a) Intrinsically stable monomers could facilitate protein oligomerization; and (b) The folded monomers could have some unknown function in the cell at small concentrations.

Another important issue is the sensitivity of the subunit stability to the point mutations used here and elsewhere. All mutants with substitutions within the subunit interface have lower stabilities than the wild-type protein [16] (Figure 3c), while these substitutions mostly destabilize the structure of subunits. For example, the replacement of Y55 by alanine resulted in a drastic decrease in the stability of hexameric Hfq [16]. Here, we show that at neutral pH, the hexameric structure of WT Hfq and its Y55W mutant cannot be unfolded by urea (Figure 3a). The triple mutation Y55W/D9A/V43R, resulting in the inhibition of protein oligomerization, also decreased the subunit resistance to urea (Figure 3b), while the subunit thermostability remained high (Figure 3d). Moreover, the residues D9, V43, and Y55 are apparently located within the region of inter-subunit contacts for the majority of bacterial Hfqs [35]. Thus, the corresponding mutations of these residues may have been the cause of the inhibition of Hfqs assembly and, hence, their function [15]. Besides, as we demonstrated in this work, the insertion of Trp residue into an inter-subunit region of oligomeric proteins may be useful to obtain information about their dissociation.

## 4. Materials and Methods

### 4.1. Reagents

To prepare buffer solutions, commercial reagents (from Sigma-Aldrich (St. Louis, MO, USA), SERVA Electrophoresis GmbH (Heidelberg, Germany), Boehringer Mannheim GmbH (Mannheim, Germany), and Invitrogen (Waltham, MA, USA)) and bidistilled water were used.

### 4.2. Proteins

The QuikChange Site-Directed Mutagenesis Kit (Stratagene, La Jolla, CA, USA) was used to introduce the mutations in the Pae Hfq protein. PCR was carried out using the pET22b(t)/Hfq plasmid [14], and the primers containing the necessary substitutions:D9A For 5′-CGCTACAAGCTCCTTACCTCAATACCCTG-3′D9A Rev 5′-CAGGGTATTGAGGTAAGGAGCTTGTAGCG-3′V43R For 5′-GAGTCTTTCGACCAGTTTCGCATCCTG-3′V43R Rev 5′-CAGGATGCGAAACTGGTCGAAAGACTC-3′Y55W For 5′-GTCAGCCAGATGGTTTGGAAGCACGCGAT-3′Y55W Rev 5′-TCGCGTGCTTCCAAACCATCTGGCTGAC-3′

The Pae Hfq Y55W mutant was purified according to the protocol developed for the purification of a wild-type protein [14]. As nothing was known preliminarily about the physical-chemical properties of Hfq triple mutant, we added to its C-end a hexa histidine-tag. This facilitates the purification of the monomeric Hfq mutant. The resulting plasmid constructs were checked by sequencing. The plasmid containing the Hfq triple D9A/V43R/Y55W mutant gene was expressed in the *E. coli* strain BL21 (DE3). The cells were grown at 37 °C in LB medium until the absorption at 600 nm reached 0.6 optical units. The plasmid expression was induced with 0.5 mM Isopropyl β-D-1-thiogalactopyranoside (IPTG). After overnight incubation at 20 °C, the cells were harvested by centrifugation at 8000× *g* for 20 min at 4 °C, resuspended in 30 mL lysis buffer (1 M NaCl, 20 mM sodium phosphate buffer, pH 6.0), and disrupted by sonication (Thermo Fisher Scientific, Waltham, MA, USA). Cell debris and ribosomes were precipitated by stepwise (successive) centrifugation at 12,000× *g* for 30 min and at 20,800× *g* for 50 min. After the addition of 20 mM imidazole, the supernatant was loaded onto an Ni-NTA agarose column (Qiagen, Düsseldorf, Germany) equilibrated with 20 mM sodium phosphate buffer, pH 6.0, 0.2 M NaCl, 20 mM imidazole. The protein was eluted with a linear gradient of imidazole from 20 mM to 250 mM. The fractions containing the target protein were concentrated and dialyzed overnight against buffer containing 50 mM Tris-HCl, pH 8.0, 200 mM NaCl.

Protein concentration was measured spectrophotometrically using molar extinction coefficients calculated according to [36], i.e., 4470 M^−1^cm^−1^ for WT Hfq and 8480 M^−1^cm^−1^ for tryptophan-containing mutants. The values of 9140 Da and 9150 Da were taken for the molecular masses of WT Hfq and its Y55W mutant subunits.

### 4.3. Crystallographic Studies

Hfq Y55W crystallization was performed at 25 °C using the hanging drop vapor diffusion method on siliconized glass in Libro plates. Drops were prepared by mixing 1–2 μL of protein solution with an equal volume of precipitant containing 7% Poly(ethylene) glycol monomethyl ether (PEG 2000 MME), 2% 2-Methyl-2,4-pentanediol (MPD), 50 mM Tris-HCl, pH 6.5. The crystals were flash cooled by liquid nitrogen after the addition of 30% glycerol as a cryoprotectant.

The X-ray diffraction data were collected at the MX beam-line 14.1 with the Pilatus 6 M detector of BESSY II (Berlin, Germany). The data were processed in XDS and scaled in AIMLESS (CCP4). The data collection statistics is presented in Table 3.

The determination of the protein structure was done by the molecular replacement technique using the Phaser program of the Phenix set [37,38]. As an initial model, the subunit of the WT Pae Hfq (PDB ID 1U1S [14]) was accepted. The asymmetric part of the crystal cell contains three protein subunits that are half of the whole hexamer, which was obtained by applying the crystallographic symmetry to this trimer. Then, the hexamer protein structure was refined using the Phenix refine program [37]. The first step included successive applications of the refinement protocols for isolated monomers (subunits) as rigid bodies, simulation of molecular annealing, and standard refinement. Then, 198 water molecules and one chloride ion were induced in the model for non-bound electron density description. The final model had good stereochemical parameters, as shown in Table 3, and was deposited in the PDB (ID 5I21).

### 4.4. Physical-Chemical Techniques

The protein absorption spectra were recorded using a Cary 100 Bio spectrophotometer (Varian Medical Systems, Palo Alto, CA, USA). The Trp fluorescence spectra were recorded at 293 nm excitation using a Cary Eclipse spectrofluorimeter (Varian Medical Systems, Palo Alto, CA, USA) with a 1 × 1 × 4 cm^3^ quartz cell. Far-UV circular dichroism spectra were measured using a Chirascan spectropolarimeter (Applied Photophysics Ltd., London, UK) with a quartz cell of 0.1 mm pathway length.

Transverse urea-gradient electrophoresis (TUGE) was performed according to a published protocol [31] with gels prepared in 20 mM Tris-HCl, pH 8.9, buffer at the 0 M–8 M urea gradient and protein run for 4 h.

The dynamic light scattering experiments were done with Zetasizer Nano ZSP (Malvern Panalytical, Malvern, UK) at 25 °C. Particle sizes were evaluated by the instrument’s software.

The size-exclusion chromatography was performed with HPLC chromatograph ProStar (Varian Medical Systems, Palo Alto, CA, USA), Superdex 200 10/300 GL column, and a flow rate of 0.4 mL/min. The calibration of the column at neutral pH was performed using various globular proteins with known molecular mass in 20 mM Tris-HCl pH 7.5, 150 mM NaCl buffer. The calibration of the column at pH 1.4 (50 mM HCl) was performed using the proteins the globularity and size of which at pH 1.4 are not essentially different from the ones at pH 7.5 accordingly to the independent data.

Non-linear fitting and smoothing of experimental data, as well as their simulations, were performed using Sigma Plot software (Systat Software Inc., Chicago, IL, USA).

### 4.5. Theoretical Basis

To reveal the changes of the protein fluorescence spectrum position caused by the denaturants within a wide range of protein concentrations, a parameter called center of mass (*CM*) was used. This parameter was calculated from each fluorescence spectrum as:(1)$$CM\left(\mathrm{nm}\right)=\frac{\sum_{i=1}^{n}\lambda_{i}\cdot A_{fl,i}}{\sum_{i=1}^{n}A_{fl,i}},$$
where *A_fl_* is the fluorescence intensity amplitude in arbitrary units at the wavelength *λ* varying between 295 and 500 nm.

This parameter seems to be less sensitive to the noise of the spectra registration, especially at low protein concentrations (in comparison, for example, with the ratio of the fluorescence intensities at the left and the right of the spectrum maximum).

To selectively reduce the high-frequency noise, the spectra were smoothed using the appropriate options of the SigmaPlot software (Systat Software Inc., Chicago, IL, USA). The smoothed spectra were deconvoluted into two or three components (depending on the unfolding model assumed) by least-square fitting using Equation (18) (see below). As suggested previously [16], the main unfolding model for HFQ should include the equilibrium transitions between the three basic states:(2)$$\frac{1}{n}N_{n}\overset{K_{NI}}{↔}I\overset{K_{IU}}{↔}U,$$
where *N*, *I*, *U* are the native, intermediate, and unfolded state of the protein subunits, respectively, while *n* is the number of subunits and
(3)$$K_{NI}^{n}=\frac{{\left[I\right]}^{n}}{\left[N_{n}\right]},$$
(4)$$K_{IU}=\frac{\left[U\right]}{\left[I\right]}.$$

With the overall polypeptide concentration (*C_m_*) introduced, and the state fractions (*f_N_*, *f_I_*, *f_U_*) as
(5)$$f_{N}=\frac{n\left[N_{n}\right]}{C_{m}},$$
(6)$$f_{I}=\frac{\left[I\right]}{C_{m}},$$
(7)$$f_{U}=\frac{\left[U\right]}{C_{m}},$$
(8)$$f_{N}+f_{I}+f_{U}=1,$$
the Equations (3) and (4) can be rewritten as
(9)$$K_{NI}^{n}=\frac{nf_{I}^{n}C_{m}^{n-1}}{f_{N}},$$
(10)$$K_{IU}=\frac{f_{U}}{f_{I}},$$

Combining the above equations, we get for the fraction of the intermediate state in the scheme (2) (equations for the two other state fractions can be written similarly):(11)$$nC_{m}^{n-1}{\left(\frac{f_{I}}{K_{NI}}\right)}^{n}+\left(1+K_{IU}\right)f_{I}-1=0.$$

Since in the case of Hfq *n* = 6, these equations can be solved only by numerical procedures. We used Newton method for finding a single real root satisfying to 0 < *f_I_ <* 1.

Depending on the solvent composition and protein concentration, the maximal population of the intermediate state can approach 1 (the two transitions are “well separated”). For example, with *K_IU_* ≈ 0, the first transition in the scheme (2) can be considered separately, with the two-state dissociation equation valid:(12)$$\frac{1}{n}N_{n}\overset{K_{NI}}{↔}I,$$
(13)$$nC_{m}^{n-1}{\left(\frac{f_{I}}{K_{NI}}\right)}^{n}+f_{I}-1=0,$$
where *f_N_ + f_I_* ≈ 1.

At any selected denaturant concentration, where Equation (13) holds and *C_m_* = *C_m_*_,0.5_ (transition midpoint, *f_N_* = *f_I_* = 0.5) we get from Equation (9):(14)lnKNI,0.5=1n[ln(n)+(n−1)ln(Cm,0.52)].

This equation was used in data analysis with fixed GuHCl concentrations (Figure 5).

To describe the unfolding of the monomeric triple mutant (see Figure 3b), a typical two-state model together with the linear extrapolation method LEM [20] was used:(15)$$I\overset{K_{IU}}{↔}U.$$

In this model, however, the state *I* displays close, but not necessarily identical, thermodynamic parameters to the intermediate state of the three-state model (3).

For the analysis of the denaturant-induced unfolding at a constant temperature, *T*_0_ = 298.15 K, the simplest LEM was used:(16)$$\Delta_{X}^{Y}G=-RT_{0}\ln\left(K_{Y,w}\right)=\Delta_{X}^{Y}G_{w}-m_{Y}C_{den},$$
where $\Delta_{X}^{Y}G_{w}$ and $\Delta_{X}^{Y}G$ are the Gibbs energy changes in water and in the presence of the denaturant, *m_Y_* is the proportionality parameter (kJ/M mol), *C_den_* is the molar denaturant concentration, and *R* is the gas constant (kJ/K mol).

Hence, to simulate the state fraction dependence on the denaturant concentration at any fixed *C_m_*, one should set *n*, *C_m_*, two proportionality coefficients, and two Gibbs energy changes in water. The dependence of state populations, and hence, fluorescence spectra, on the protein concentration can be used for estimating the spectrum of the intermediate state at a denaturant concentration where the population of the unfolded state is negligible, and the fraction of the intermediate state can be raised to 1 by protein dilution. In this case, tryptophan fluorescence could be the most useful, as spectra could be reliably measured at very low protein concentrations (in our case, down to 0.4 µM).

To change the population of various conformational states during the equilibrium protein unfolding, we did the following.

For high-frequency noise reduction, the spectra, particularly those recorded at low protein concentrations (<0.05 mg/mL), were smoothed using the appropriate options in the SigmaPlot software (Systat Software Inc., Chicago, IL, USA). In some cases, the spectra were standardized by dividing them by the intensity at some selected wavelength. Such standardization minimizes the error caused by protein concentration uncertainty. The spectra were deconvoluted into two or three components (two- or three-state models) using the least-square fitting based on the following equations:(17)$$Fl_{sym}\left(\lambda \right)=f_{N}fl_{N}\left(\lambda \right)+f_{I}fl_{I}\left(\lambda \right)+f_{U}fl_{U}\left(\lambda \right),$$
where *f*_X_ and *fl*_X_ are the populations and basic fluorescence spectra of the corresponding states. Three basic spectra (for the native, monomeric, and unfolded state) were estimated as described above (Figure 2c) and assumed to be independent of buffer composition. In the case of a negligible unfolded state population, this equation transforms into:(18)$$Fl_{sym}\left(\lambda \right)=f_{N}fl_{N}\left(\lambda \right)+f_{I}fl_{I}\left(\lambda \right).$$

## Figures and Tables

**Figure 1 molecules-27-03821-f001:**
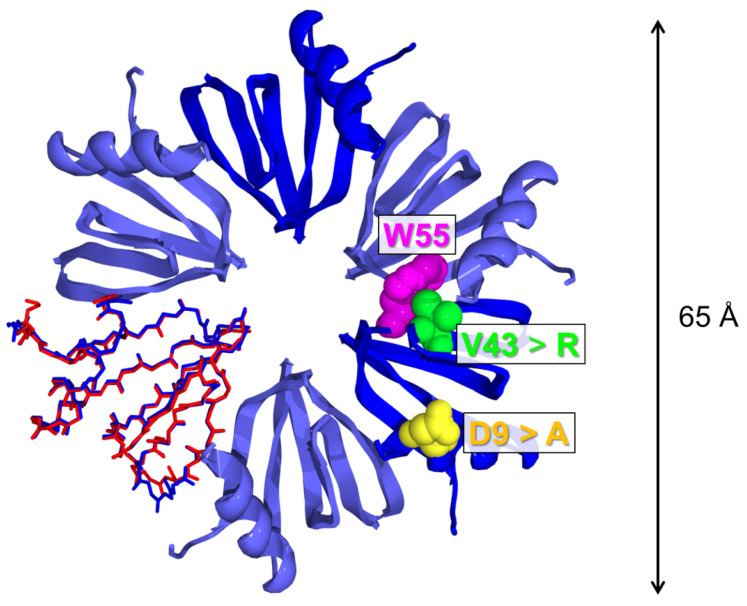
The crystal structure of the Hfq Y55W mutant: the tryptophan residue (W55) inserted into the hexameric structure and additional mutations (V43R, D9A) serving to obtain the protein monomeric form are shown in rectangles. One subunit represents the superposition of the subunit main chain conformations for the WT (red, PDB code 1U1T) and Y55W (blue, PDB code 5I21) Hfq variants. The double arrow to the right of the figure shows the external diameter of the protein ring.

**Figure 2 molecules-27-03821-f002:**
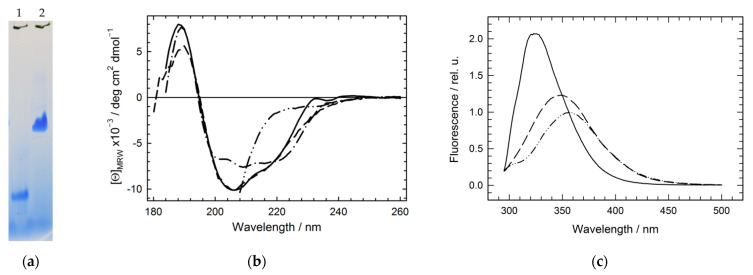
Structural properties of the hexameric Hfq Y55W and monomeric Hfq Y55W/D9A/V43R mutants: (**a**) Native polyacrylamide gel electrophoresis (PAGE) of the monomeric (track 1) and hexameric (track 2) forms; (**b**) Far-UV circular dichroism spectra for WT (dash-dot line), Y55W (solid line), Y55W/D9A/V43R (dashed line) Hfq variants, and any variant in the presence of 5 M GuHCl (dash-dot-dot line); (**c**) The fluorescence emission spectra for Hfq Y55W in three states (Equation (2)): the native hexameric (solid line, spectrum center of mass CM = 337 nm), the monomeric Y55W/D9A/V43R mutant(dashed line, CM = 355 nm), and unfolded monomeric (in the presence of 5 M GuHCl; dash-dot-dot line, CM = 360 nm); Protein concentration (monomer): 65 µM; Buffer: 20 mM Tris-HCl, pH 8.5.

**Figure 3 molecules-27-03821-f003:**
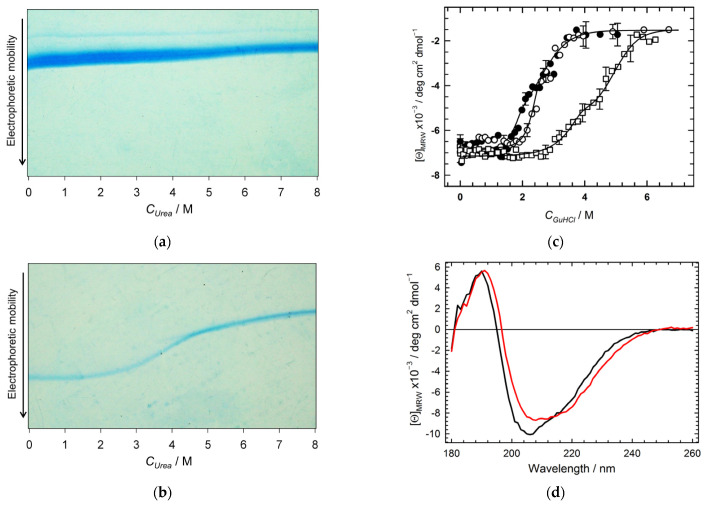
The effect of urea on Hfq structure (**a**,**b**), GuHCl (**c**), and temperature (**d**): (**a**) Transverse urea gradient gel electrophoresis of the Hfq Y55W hexamer; (**b**) Transverse urea gradient gel electrophoresis of the Hfq Y55W/D9A/V43R monomer; (**c**) The CD-monitored GuHCl-induced unfolding of the Hfq Y55W (circles) at 66 μM (open symbols) and 6 μM (filled symbols), and Hfq WT (squares) at 66 μM. The error bars correspond to the deviations from average values as a result of five measurements. The solid lines stand for the best fittings of the data with the parameters shown in Table 2; (**d**) The CD spectra of monomeric triple Hfq mutant at 20 °C (black line) and at 90 °C (red line) at a protein concentration of 1.24 mg/mL (124 μM of monomers).

**Figure 4 molecules-27-03821-f004:**
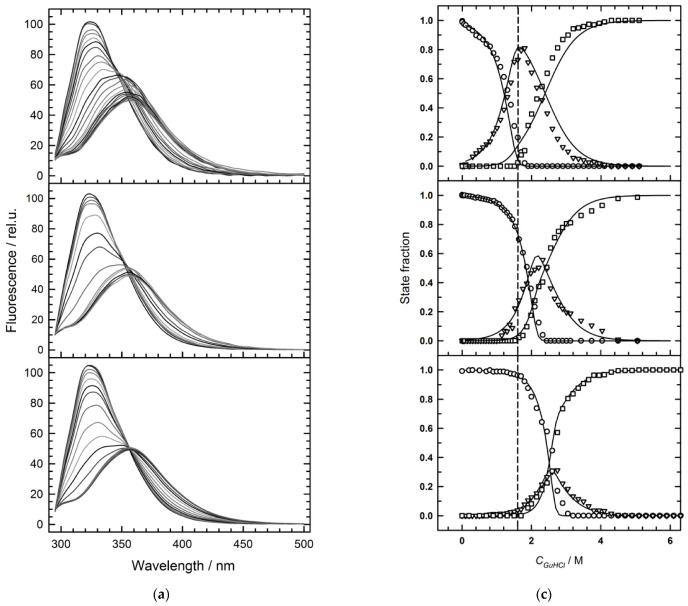
Analysis of the GuHCl-induced unfolding at various Hfq Y55W concentrations: (**a**) A set of selected Y55W fluorescence spectra recorded within the 0 ÷ 5 M GuHCl range (from top to bottom, grayscale) at three protein (monomer) concentrations: 0.8 (top), 6.5 (middle), and 65 µM (bottom); (**b**) The GuHCl dependence of the spectrum mass center values (Equation (1)) for three Hfq Y55W concentrations. Circles stand for 0.8 µM, squares for 6.5 µM, and triangles for 65 µM. The black symbols represent results of refolding of Hfq Y55W; (**c**) The GuHCl concentration dependences of the state fractions, as derived from the deconvolution of Y55W fluorescence spectra at three protein concentrations: 0.8 (top), 6.5 (middle), and 65 µM (bottom). Symbols show: hexamer, circles; intermediate, triangles; and unfolded monomer, squares; while solid lines represent simulations based on the three-state unfolding model (3), Equations (3)–(11), and the parameters from Table 2. The vertical dashed line indicates a GuHCl concentration of 1.6 M; (**d**) The result of the best fit of the fluorescence spectrum of Hfq Y55W (6.5 µM) at 1.98 M GuHCl (black solid line, experiment; red dashed line, the best fit to the three-state model). State contributions: hexamer, blue solid line; intermediate monomer, dark green dashed line; unfolded monomer, dark pink dash-dot-dot line. Best fit fractions: f_n_ = 0.36, f_i_ = 0.47, f_u_ = 0.17.

**Figure 5 molecules-27-03821-f005:**
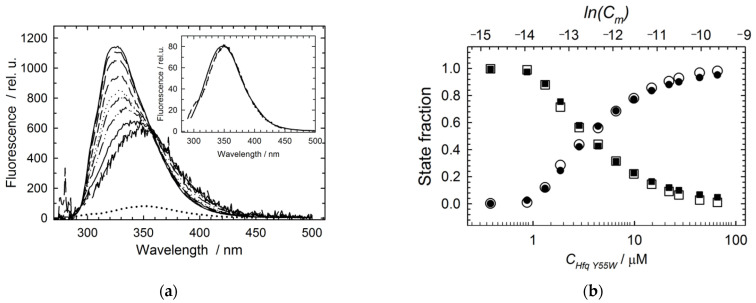
Disassembly of Hfq Y55W in the presence of 1.6 M GuHCl by decreasing protein concentration: (**a**) The set of selected fluorescence spectra of Hfq Y55W at various protein (monomer) concentrations (from the top line to the bottom line—65 µM, 22 µM, 10 µM, 6.5 µM, 4.4 µM, 2.8 µM, 1.9 µM, 1.3 µM, 0.87 µM, 0.38 µM). The spectra are standardized to the same concentration (65 µM). At the inset: the basic spectrum of the triple mutant (solid line) assumed in fluorescence spectra deconvolution, and the spectrum of the intermediate, obtained by the dilution down to 0.38 µM at 1.6 M GuHCl (dashed line); (**b**) The dependence of the state fractions on the molar protein (monomer) concentration determined by deconvolution of the fluorescence spectra. The fractions of the native (hexameric) and intermediate states are shown by filled circles and filled squares, respectively. The open symbols are the prediction of the state distribution obtained from simulations based on the parameters from Table 2. The buffer contains 20 mM Tris-HCl, pH 7.5, and 1.6 M GuHCl.

**Figure 6 molecules-27-03821-f006:**
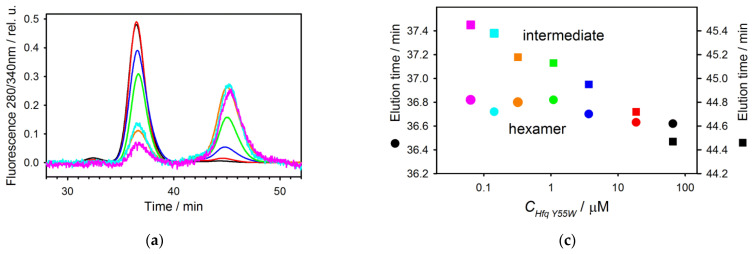
The size-exclusion chromatography of Hfq Y55W at 1.6 M GuHCl: (**a**) The set of standardized elution profiles of Hfq Y55W at various concentrations of the protein loaded (67 µM is shown in black, 18.9 µM—in red, 3.7 µM—in dark blue, 1.1 µM—in green, 0.33 µM—in yellow, 0.15 µM—in cyan, and 0.064 µM—in pink); (**b**) The dependence of the elution time on the protein molecular mass (the chromatographic column calibration); proteins used for calibration are aldolase (158 kDa), bovine serum albumin (66 kDa), bovine carbonic anhydrase B (29 kDa), cytochrome *c* (12.3 kDa). The arrows indicate the elution times of the respective peaks at the highest (black) and the lowest (pink) concentration of the protein loaded; (**c**) The dependence of the elution peak time on the protein concentration loaded (◯—initial peak, ◻—peak of the dissociated intermediate; the color coding is the same); (**d**) The dependence of the protein state populations (determined as the areas of corresponding elution peaks) on the protein concentration loaded (◯—hexamer, ◻—dissociated intermediate; color coding is the same).

**Figure 7 molecules-27-03821-f007:**
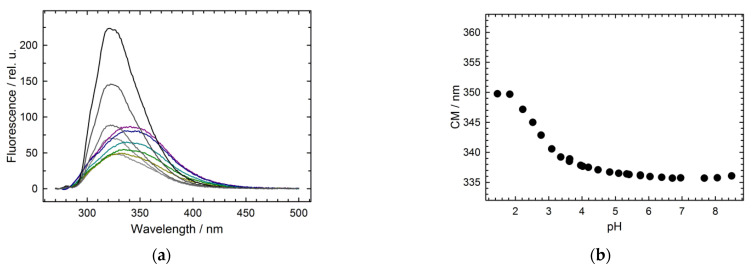
The pH dependence of Hfq Y55W fluorescence spectrum. (**a**) Fluorescence spectra measured at pH 6.0, 5.07, 4.15, 3.53, 3.03 (lines range from black to light grey); 2.72, 2.16, 1.76, 1.6, 1.4 (dark yellow, green, light blue, dark blue, and purple lines, respectively). Protein (monomer) concentration is 67 μM. (**b**) The pH dependence of the spectrum center of mass (CM). For different pH values, the following buffer solutions were used: 20 mM Tris-HCl, 30 mM NaCl for pH 7.6–8.5; 20 mM Na-Cacodylate-HCl, 30 mM NaCl for pH 5.35–6.93; 20 mM Na-Acetate, 30 mM NaCl for pH 3.55–5.65; 20 mM Glycine-HCl, 30 mM NaCl for pH 2.16–3.83; 20 mM HCl, 30 mM NaCl for pH 1.76; 50 mM HCl for pH 1.4.

**Figure 8 molecules-27-03821-f008:**
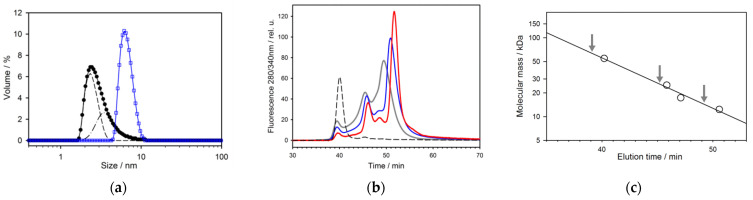
The hydrodynamic properties of pH-induced intermediates of Hfq Y55W. (**a**) Size distribution of Hfq Y55W at pH 7.6 or WT Hfq at pH 1.4 (blue squares) and that of Hfq Y55W at pH 1.4 (black circles) revealed by dynamic light scattering at a protein (monomer) concentration of 100 µM. The deconvolution of the mutant peak at pH 1.4 into peaks corresponding to 35 Å (dashed-dot line) and 23 Å (dashed line); (**b**) The size-exclusion chromatography of Hfq Y55W on Superdex 200 10/300 GL at pH 1.4 and various applied protein concentrations: the grey line stands for 3.0 mg/mL (300 μM monomer); the blue line for 0.6 mg/mL (60 μM monomer); the red line for 0.12 mg/mL (12 μM monomer). The dashed line shows the hexameric WT Hfq elution profile at pH 1.4; (**c**) The chromatographic column calibration. Proteins used for calibration are WT Hfq (54.6 kDa), chymotrypsinogen (25 kDa), myoglobin (17.3 kDa), cytochrome *c* (12.3 kDa). The arrows indicate the elution times of the peaks at 300 µM protein concentration loaded.

**Table 1 molecules-27-03821-t001:** The secondary structure content (%) of Hfq variants as shown by far-UV CD spectra deconvolution [30].

Protein	α-Helix	β-Structure	Others
WT Hfq (hexamer)	10.8	41.0	48.2
Hfq Y55W (hexamer)	8.4	39.4	52.2
Hfq Y55W/D9A/V43R (monomer)	10.5	30.9	58.6

**Table 2 molecules-27-03821-t002:** The thermodynamic parameters of Hfq equilibrium unfolding transitions (as shown by the linear extrapolation method (LEM) [20,21]).

HFQ Variant	Denaturant	Method	Model	*m_NI_* (kJ/mol)	*K_NI_*_,*w*_ (mol^5/6^)	*m_IU_* (kJ/mol)	*K_IU_* _,*w*_	Δ*_I_^U^G_w_* (kJ/mol)
WT	GuHCl	CD	Equations (2) and (11)	5.9	1.5 × 10^−7^	5.5	2.8 × 10^−5^	−26.0
Y55W (hexamer)	GuHCl	Flu.	Equations (2) and (11)	6.2	3.5 × 10^−7^	5.5	3.5 × 10^−3^	−14.0
	GuHCl	CD	Equations (2) and (11)	6.2	3.5 × 10^−7^	5.5	3.5 × 10^−3^	−14.0
	dilution at 1.6 M GuHCl	Flu.	Equations (12) and (13)	6.2	4.0 × 10^−7^	-	-	-
Triple mutant	Urea	TUGE *	Equations (15) and (18)	-		3.9	4.1 × 10^−3^	−13.3

* The data were derived from Figure 3b.

**Table 3 molecules-27-03821-t003:** X-ray data parameters and refinement statistics for the Pae Hfq Y55W mutant. Statistics for the highest-resolution shell is shown in parentheses.

PDB ID	5I21
Wavelength (Å)	0.86
Resolution range (Å)	37.43–1.55 (1.605–1.55)
Space group	*P*6
Unit cell	114.4 Å, 114.4 Å, 28.3 Å, 90°, 90°, 120°
Total reflections	242,835 (23,797)
Unique reflections	31,293 (3038)
Multiplicity	7.8 (7.8)
Completeness (%)	1.00 (0.99)
Mean *I*/sigma (*I*)	25.89 (4.09)
Wilson *B*-factor	16.57
R-merge	0.04633 (0.5224)
R-meas	0.04967 (0.5593)
CC1/2	1 (0.908)
CC*	1 (0.976)
R-work	0.1877 (0.2138)
R-free	0.2135 (0.2418)
CC(work)	0.965 (0.892)
CC(free)	0.943 (0.880)
Number of non-hydrogen atoms	1791
macromolecules	1592
water molecules	198
ions (Cl-)	1
RMS (bonds)	0.006
RMS (angles)	0.84
Ramachandran favored (%)	97
Ramachandran allowed (%)	3.1
Ramachandran outliers (%)	0
Rotamer outliers (%)	1.1
Clashscore	3.96
Average B-factor (Å^2^)	23.60
macromolecules	22.22
ligands	40.51
solvent	34.66

## Data Availability

Structural data are available in the Protein Data Bank (PDB) database under the accession numbers 5I21 (http://doi.org/10.2210/pdb5I21/pdb, accessed on 13 April 2022).

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
