# Peer review of "The Denaturant- and Mutation-Induced Disassembly of Pseudomonas aeruginosa Hexameric Hfq Y55W Mutant"

_molecules, 2022, doi:10.3390/molecules27123821_

Round 1

Reviewer 1 Report

In this work the authors analyzed the stability of a homohexameric protein towards temperature and chemical denaturants, with the aim to contribute to a better understanding of protein folding. In particular, the role of the quaternary structure for protein stability was aimed to be addressed in this study.

Major comments

  • While the study is technically sound, it lacks a proper discussion, in which it is described what new information about protein stability is gained from the current study. There are numerous studies where it was shown that quaternary structure formation increased chemical and thermal stability of proteins. Thus, this result per se is not new.
  • At least the results could be discussed in the context of the proteins function.
    For example, are there Hfq variants from other species, which are less stable than the one from P.aeruginosa? Are there cases where amino acids like the ones introduced in the current studies are located at comparable positions? Is dissociation of the protein in any context an issue in vivo?
  • Something is wrong with the CD-spectra in Fig.2. There should be three different ones, but there seem to be four. Furthermore, does it make sense, that the spectra are “reliably different” (line 99) but the secondary structure prediction is very similar (Tab. 1) ? Also to my knowledge a particular contribution of trp in the wavelength region as hypothesized in line 102 is not to be expected.

Furthermore, some details in the data analysis should be explained a more clearly (see below).

Minor comments

  • The function of the protein, in particular with respect to its structure, should be described in more detail in the introduction.
  • Line 39: half transition temperature
  • For the x-ray structure a reference is given (20), which I cannot find. In the pdb-data bank its described as “to be published”.
  • Lin 63: “the introduction of a tryptophan……is aimed..” Grammatically wrong, better: “the aim of introducing a tryptophan….”
  • Line 74: “nearest neighbour” is a bit misleading in this context, I would rather say “sensitive to its surrounding “
  • Line 121: “somewhat worse” should be replaced by “more pronounced “
  • Line 134: was reversibility of unfolding in urea shown only for the monomeric mutant? What about the y55w mutant, and unfolding with GdmHCl? Only for reversible unfolding the numbers obtained in the thermodynamic analysis are meaningful.
  • In Fig. 4c I see only three curves instead of four. If there is only one data set for the wt protein, the text in the figure legend should be adjusted.
  • It is not entirely clear, how the parameter in tab. 2 were obtained: were the data from both concentrations of the y55w variant fitted simultaneously? Was “miu=5.5” set constant for all? It seema to be unlikely that exactly the same value is found without any constraints.
  • Line 160: the ref. to figure 2a is confusing here.
  • Line 167: “therefore we used some thermodynamic parameters…”. Used for what? The paragraph describes why thermal denaturation data could not be be analyzed properly; it is unclear how results from the chemical denaturation can help here.
  • Line 182: down to 0.01 µM?
  • Line 184: the meaning of the sentence is not entirely clear. Do you mean, that since there is obviously not a single isosbestic point, there must be more than two states (=more than one transition?)
  • For the analysis of the denaturation observed via trp-fluorescence, three basis spectra were fitted to the spectra obtained at different GdmHCl concentrations. The basis spectra employed are the hexameric one (y55w), the monomeric one (triple mutant) and the one at 5 M GdmHCl. Thus, it is assumed here, that the additional two substitutions have no influence on the fluorescence spectrum. This should be stated somewhere. Furthermore, in particular for the conditions, where the intermediate (= folded monomer) is significantly populated, the goodness of the fit (deconvolution) should be demonstrated by an overlay of the measured and the calculated spectrum. In fact it is stated in line 192, “it is clear that the deconvolution quality is acceptable…”, but no information which supports this statement is given.
  • Line 220: strictly speaking, based on the position of the CM at 357 nm one cannot confirm that the corresponding species is monomeric; therefore I would rather say, “as one would expect for the monomeric species”
  • 5: in the methods-section it is stated that for analysis of the fractional distribution eq. 15 is used; in the table it is 13,14. Furthermore, how eq. 15 was used should be explained better. What is KIU,0.5? KIU should be independent on protein concentration…
  • Axis in fig. 5b: the x-axis should be presented the same way as in fig. 6, for clarity.
  • Line 232: the open symbols are the prediction of the state distribution based on the parameters obtained from the previous analysis (parameters from Tab.2), correct? This is not entirely clear from the figure legend.
  • Size exclusion data: this are obtained in presence of 150 mM NaCl, the rest of the experiments were obtained in absence NaCl. This should be pointed out, and the potential consequences of different salt concentrations discussed.
  • DLS measurements (Fig. 8): where the distribution for Hfq y55w at pH 7.6 and wt at pH 1.4 really identical? Usually in DLS measurements even two distributions of the same sample are rarely identical…Why was the volume representation chosen?
  • Line 308: what is meant by “opens one more intermediate peak”?
  • Line 311: shouldn’t the fraction of trimers and dimers also change with protein concentration?
  • Line 320: “the high intrinsic stability of the subunit implies..”. It is the other way round: the observed population of the monomeric intermediate indicates that the subunits are stable.
  • Line 335 ff: the relation between the stability monomers, enthalpy gain and analysis of DSC experiments is not clear.
  • Line 347: sufficiently high for what?
  • Line 364: is only the monomer isolated via his-tag?
  • 1: the necessity of doing this normalizisation is not evident, and it is not clear when it is used. For the deconvolution it is not required; here one can always express the fitted fractions relative to the sum of fractions, so normalization for the protein concentration is not necessary.
  • Line 477: the meaning of “one should set….” is not clear. Do the authors mean: these parameters are varied to obtained the best fit?
  • Line 479: in the analysis (see above) its assumed that the spectrum of the triple mutant can be used as representant of the monomer species in the denaturation of the y55w mutant. Here it is mentioned that the spectrum of the intermediate is estimated. This seems to be a contradiction in the approach. Furthermore, the paragraph is not very clear; so how the spectrum is obtained is difficult to understand.
  • Line 490: here again it is mentioned that spectra are standardized based on the isobestic points; see comment above.
  • It seems that spectra where smoothed in some cases. Did the authors check that the shape of the spectra where not altered? One of the CD-spectra in Fig. 2 looks a bit strange.

Reviewer 2 Report

This is a classic and well-developed analysis of protein unfolding. By applying manly spectroscopic methodologies the authors analyze the unfolding of an hexameric protein. This is precisely the most relevant part of this study, since few unfolding/refolding studies have been done with multimeric proteins.

The authors indicate (lines 132-134) that the unfolding is fully reversible. Possibly some experimental data indicating this fact could be added either in the current form of the manuscript or, perhaps more adequately, in additional information. Reversible refolding might be more relevant to the in vivo process of protein synthesis.

However, the current from of the manuscript needs only minor changes:

It is not clear why the transverse urea gradient shows that the triple mutant shows a cooperative urea induced transition (line 130).

Also the word "Note" in lines 128 and 132 are not needed, since the data is not shown, we can not note anything.

Reviewer 3 Report

The authors have employed a range of complementary techniques to study the dissociation of the hexameric Hfq protein in their study, presented in a well-structured manuscript. However, the evidence for the existence of the intermediate dimer/trimer state during disassembly at neutral pH is not currently very convincing. The only direct observation presented here is via the use of size exclusion chromatography, which does not show conclusively that the non-hexamer peak corresponds to such an intermediate, rather than monomer. This is in stark contrast to the low pH data, which show three peaks, clearly indicating the presence of some species in addition to the monomeric and hexameric forms. The indirect evidence, namely fitting of dissociation curves, is not conclusive in its present form; the only model considered is one with three-states, with no comparison to a two-state model. Without such a comparison, the conclusions are not convincing to the reader.

Specifically:

Major points which must be addressed before publication:

  1. Several datasets throughout the paper which would strengthen the conclusions are not shown, these need to be included in a supplementary information file if they cannot fit in the main text.
  2. Figure 3c: It is not clear how strongly these data point to a 3-state model.
    1. Include error bars for the data so one can judge how significant deviations from the fitted model/s are.
    2. Include the best fit to a 2-state model, to compare directly with the 3-state model.
  3. Figure 4c: The authors should state which sample was used to provide the fluorescence spectrum for the intermediate state, and an explanation of why this is an appropriate choice to represent the intermediate state.
  4. Figure 6: The authors need to include control data to demonstrate the elution time of the monomer. This is a crucial control to confirm whether the peak at elution time ~45 minutes is monomer or an intermediate state. (Note that the elution times cannot be directly compared to those in fig 8 at low pH, demonstrated by the hexamer peak which is ~37 mins at neutral pH but ~40 at low pH.)

Minor points which would improve the paper quality:

  1. Please elaborate on the choice of the specific mutations studied.
  2. Lines 83-4: Which small changes? Please expand.
  3. Throughout the paper, ‘centre of mass’ is used to monitor spectral shifts in fluorescence properties, please expand on why is this preferable to simply taking the wavelength of maximum emission.
  4. Table 2: State which figure/panel was used to calculate each row, and use subscripts where appropriate in the column headings.
  5. Line 175: eleven-fold?
  6. Lines 184-186: It is unclear whether these are really two quasi-isosbestic points or is just one with some level of experimental noise. Please expand on this in the text.
  7. A few general language edits: ‘conservative’→‘conserved’, ‘besides’→‘moreover’.
  8. Figure 2a: If possible, include a marker lane to have some approximation of the protein size.
  9. Figure 2b/c: Please make the linestyles consistent between the two plots to make the figure easier to understand. Currently, the same sample (that with GuHCl) is represented by different linestyles in the two plots.
  10. Figure 2c: Please clarify either in the plot or caption on whether the y-axis represents the fluorescence emission spectra.
  11. Figure 3a/b: If possible, provide a comparison of the electrophoretic mobilities and/or apparent sizes with the gels in figure 2a.
  12. Figure 4a: Change to a monotonic colour scheme to make the plot clearer.
  13. Figure 4b: Please include the raw data (as in 4a) for all concentrations in the supplementary information.
  14. Figure 5a: Please include the protein concentrations studied in the caption.
  15. Figure 6: The colour scheme is difficult to follow, the yellow points and lines are particularly difficult to see.
  16. Figure 6c: This calibration assumes that the Hfq species are all globular, is this true for the dissociated/denatured states?
  17. Figure 6: Subpanels are labelled a/b/c/d differently from in previous figures
  18. Figure 7a: Please expand on why the fluorescence decreases and then increases with decreasing pH.
  19. Different buffers were used for the different pH ranges. Please provide evidence to show that the different ions do not have additional effects on the protein, other than that of the pH itself.

Round 2

Reviewer 1 Report

The authors adressed my concerns in a satisfactorily way.

Only two minor points:

Line 174: as expected from a process involving dissociation of an oligomer (not a general three state process).

Fig. 4b: the reversible points in dark grey are difficult to see. Maybe red? Or black?

Author Response

Response to Reviewer 1 Comments

Dear Reviewer,

Thank you for your help in improving our article.

Point 1: Line 174: as expected from a process involving dissociation of an oligomer (not a general three state process).

Response 1: We corrected the text (see lines 176-177).

Point 2: Fig. 4b: the reversible points in dark grey are difficult to see. Maybe red? Or black?

Response 2: The reversible points on Fig. 4b are black now.

Sincerely yours,

Gennady V. Semisotnov,

Vladimir V. Filimonov,

Corresponding authors

Reviewer 3 Report

Thank you for addressing many of the points raised. However, the authors have not addressed several of my points required to fully support the conclusions, most importantly it is crucial to include all data referenced in the text. There is no reason not to include such data in the supplementary information, especially given that many journals require the raw data to be made available, not only to be presented in supplementary files.

Author Response

Response to Reviewer 3 Comments

Dear Reviewer,

Thank you for your help in improving our article.

Point 1: Thank you for addressing many of the points raised. However, the authors have not addressed several of my points required to fully support the conclusions, most importantly it is crucial to include all data referenced in the text. There is no reason not to include such data in the supplementary information, especially given that many journals require the raw data to be made available, not only to be presented in supplementary files.

Response 1: We included in Figure 4a the additional sets of the protein fluorescence spectra for the protein concentrations of 0.8 µM and 65 µM. Moreover, we included (as a Supplementary File) raw far-UV CD spectra selected for the crucial points of GuHCl-induced unfolding of the proteins presented in Figure 3c.

Sincerely yours,

Gennady V. Semisotnov,

Vladimir V. Filimonov,

Corresponding authors